# Chronic inflammation was a major predictor and determinant factor of anemia in lactating women in Sidama zone southern Ethiopia: A cross-sectional study

**Tafere Gebreegziabher**[1]*, **Taylor Roice**[1], **Barbara J. Stoecker**[2]

**1** Department of Health Sciences, Central Washington University, Ellensburg, WA, United States of America,
**2** Department of Nutritional Sciences, Oklahoma State University, Stillwater, OK, United States of America

* tafere.bl@gmail.com

**Data Availability Statement:** All relevant data are within the manuscript and its supporting information files.

## Abstract

Anemia in women of reproductive age is highly prevalent globally and remains a public health problem. In Ethiopia, despite efforts to minimize the burden of anemia, it is still a moderate public health problem. Anemia has various etiologies including nutritional deficiency, parasitic infection, and inflammation. The aim of this study was to examine contributing factors to anemia in lactating women. Following ethical approval, and six months after delivery, all lactating women (n = 150) were recruited to participate in this study from eight randomly selected rural villages. Anthropometric and socio-economic factors were assessed. From each, a blood sample was collected for measuring hemoglobin, iron biomarkers, zinc, selenium, and inflammation markers. The median (IQR) hemoglobin (Hb) was 132 (123, 139) g/L. Of the women, 19% were anemic and 7% had iron deficiency anemia; 31% were iron deficient and 2% had iron overload. Also, 8% had functional iron deficit, 6% had acute inflammation, 13% had chronic inflammation, and 16% had tissue iron deficiency. The majority (78%) of the women had low plasma zinc out of which more than 16% were anemic. Hb was positively associated with plasma iron and plasma zinc and negatively associated with transferrin receptor (TfR) and α-1-acid glycoprotein (AGP). Plasma iron, AGP, TfR, hepcidin and plasma zinc were significant predictors of maternal anemia. Additionally MUAC and level of education were associated positively with maternal hemoglobin. This study showed that maternal anemia was associated with multiple factors including nutritional deficiencies, inflammation and limited education.

## Introduction

Anemia is characterized by a decline in the number and size of red blood cells that results in insufficient oxygen carrying capacity to meet physiological needs [1]. Globally, anemia in women of reproductive age (WRA) (15 to 49 years old) remains a public health problem and there were more than 528 million anemic WRA in 2011 out of which 496 million were non-pregnant women [2]. Prevalence of anemia in non-pregnant women was nearly 11% higher in

**Funding:** The research was funded by the USDA Multistate Project, W-3002 to BJS; Nestlé Foundation to TG. The funders had no role in study design, data collection and analysis, decision to publish, or preparation of the manuscript.

**Competing interests:** The authors have declared that no competing interests exist.

2011 than in 1995 [3]. In Ethiopia, the progress towards reducing prevalence of anemia seems erratic. In 2005, the percentage of WRA reported to be anemic was 27% and in 2011 this figure had declined to 17%. However, in 2016 the percentage was reported to be 24% and was classified as a moderate public health problem [4]. Anemia results in reduced work productivity which could be due to reduced oxygen carrying capacity in an individual's blood [5].

Anemia has various potential etiologies. Iron deficiency has been considered the major cause of anemia and contributes to approximately 50% of all anemia worldwide [3, 6], but other micronutrient deficiencies (including vitamin A, folate, zinc and vitamin $B_{12}$), parasitic infection, and inflammation can cause anemia as well [1, 6, 7]. Numerous determinant factors of anemia have been reported in multiple geographic settings. These include infection, lack of bioavailable dietary iron, being overweight, low education level, unemployment, seasonal variation, and other socio-economic and demographic factors [8–11]. According to UNICEF, the underlying causes of anemia are household food insecurity, inadequate care, unhealthy household environment, and lack of health services [12]. Although some factors mentioned may not cause anemia directly, all are interrelated. For instance, low household income leads to poor diet and poor health services. A combination of these factors increases risk of nutritional deficiencies, impaired immunity, infection, and inflammation which ultimately lead to anemia [13].

To accurately assess micronutrients and their role in anemia has been a challenge. In the study of factors contributing to anemia, it is imperative to assess nutritional biomarkers such as ferritin, transferrin receptor, zinc, and others. However, relying on these biomarkers alone to determine iron status could be misleading because they can be substantially overestimated or underestimated in the presence of inflammation or malaria infection [14]. As a result, these biomarkers should be adjusted for inflammation before interpretation of results, particularly in settings with high infectious disease burden [13–15].

Repeated and extensive surveys since 2012 have examined associations between nutrient biomarkers, inflammation and anemia [16]. Iron and inflammatory biomarkers were consistently associated with anemia in children as well as in WRA [6, 17]; more importantly micronutrient deficiency combined with infection makes anemia more prevalent. Ethiopia is one of the countries where burden of malaria and intestinal parasitic infection are quite high, particularly in rural areas of the southern region [18]. In such places it is paramount to measure infection biomarkers in order to determine contributing factors to anemia. Hence, the aim of this study was to examine determinant factors of anemia in lactating women from Sidama zone, southern Ethiopia, focusing on iron status and infection biomarkers.

## Materials and methods

### Study population and design

We conducted a cross-sectional study of lactating Ethiopian women from eight randomly selected rural villages in Sidama zone, southern Ethiopia. In June to August, 2013, all women 6 month post-delivery were invited to participate in the study. Eligibility criteria were: the woman must be 18 years of age or older, must be lactating six months after delivery and must have no history of illness. Because all women agreed to join the study, we consider the study representative of the rural Sidama population. Women came to their local community health post for the data collections. The study period was the rainy season and most families in the area lived from subsistence farming with enset (false banana) and maize as their staple crops.

Malaria infection varies geographically and seasonally, but the malaria epidemic is normally severe during the rainy season when temperature is high [19]. The study area which is known

for high incidence of malaria every year is relatively hot during the rainy season which creates a suitable environment for malaria outbreaks.

## Ethical clearance

Ethical approval was given by the review boards for Hawassa University and the Ministry of Science and Technology, Ethiopia. Informed consent was signed by each woman.

## Anthropometry and socio-economic characteristics of women

Weight in light clothing was measured to the nearest 100 grams using a solar digital scale (Uni-scale, UNICEF, NY, USA). Height was measured to the nearest 0.1 cm using a single calibrated instrument (Adult Board, Schorr Productions, Olney, MD, USA). Mid upper arm circumference was measured to the nearest 0.1 cm using a plastic measuring tape. A questionnaire was administered individually to assess demographic and socio-economic characteristics of the women. Principal component analysis (PCA) was applied to compute wealth index, and the score was used to divide the participants into five quintiles. The household Food Insecurity Access Scale (HFIAS) developed by the Food and Nutrition Technical Assistance (FANTA) project was used to assess food insecurity [20]. The nine questions ranged from simple worry for food shortage to the experience of often spending day and night without food during the prior four weeks. The scale was computed into four levels of food insecurity including: food secure, mildly food insecure, moderately food insecure, and severely food insecure.

## Laboratory methods

A morning venipuncture blood sample was collected from each participant using a disposable 10 cc syringe coated with lithium heparin with a 21 gauge needle (Sarstedt, Inc., Newton, N. C.). A drop of venous blood from the syringe needle was used for hemoglobin measurement. Hemoglobin concentration was measured at the health post with a Hemo-Cue (Hemocue AB, Ängelholm, Sweden) instrument. The remaining blood was centrifuged and plasma was separated immediately. Plasma was frozen at– 20°C and used for measurement of selected minerals and inflammation biomarkers. According to WHO, anemia in non-pregnant WRA is defined as Hb concentration $< 120$ g/L. Moreover, anemia was further classified as mild (Hb 110–119 g/L), moderate (Hb 80–109) and severe (Hb $< 80$ g/L) [1].

Plasma iron, zinc, and selenium were measured by inductively coupled plasma mass spectrometer (ICP-MS, Elan 9000, Perkin Elmer, Norwalk, CT, USA) using UTAK serum (Utak Laboratories, Inc., Valencia, CA, USA) for quality control. Plasma ferritin and transferrin receptor (TfR) were quantified using an ELISA procedure (Ramco Laboratories, Stanford, TX, USA). Hepcidin-25 was quantified using ELISA (DRG Inc., Mountainside, NJ, USA). ELISA kits also were used to assess C-reactive protein (CRP) (Helica Biosystems, Inc., Fullerton, CA, USA) and α-1-acid glycoprotein (AGP) (R & D Systems, Inc., Minneapolis, MN, USA).

Hemoglobin was adjusted for altitude according to the equation recommended by UNICEF/UNU/WHO: Hb (g/dL) = -0.32 x (altitude in meters x 0.0033) + 0.22 x (altitude in meters x 0.0033)$^2$ [21, 22]. CRP was classified as high (acute inflammation) if it was >5 mg/L. Values greater than 1 g/L for AGP were taken to represent chronic inflammation. Prior to defining iron deficiency and iron deficiency anemia, ferritin concentration and transferrin receptor were adjusted for inflammation using the formula specified by the Biomarkers Reflecting Inflammation and Nutritional Determinants of Anemia (BRINDA) team [15, 23] as follows:

Exp (unadjusted ln biomarkers–$\beta1$ ($CRP_{observed}$ − maximum of lowest decile for CRP) − $\beta2$ ($AGP_{observed}$ − maximum of lowest decile for AGP)).

Iron deficiency was defined as adjusted ferritin < 15 µg/L, and iron deficiency anemia was defined as iron deficiency concurrent with anemia. Concentration of sTfR ≥ 8.3 mg/L was taken to represent a functional iron deficit. Body iron was calculated as recommended by Cook et al. [24, 25] as follows: Body iron (mg/kg) = —[$\log_{10}$ (sTfR/F ratio)– 2.8229] / 0.1207.

Zinc inadequacy was defined as plasma zinc < 10.7 µmol/L [26, 27]. No universal interpretive criterion has been set for plasma selenium because selenium notably varies with geographic location [28].

## Statistical analysis

Data were analyzed using SPSS, version 23 (SPSS Statistics Version 23, IBM Corp., Armonk, NY, USA). All skewed data including iron and inflammation biomarkers were log transformed before analysis. Percentages, means, standard deviations, medians, and interquartile ranges were used as appropriate in describing the socio-economic and demographic characteristics as well as the concentrations of minerals and inflammatory markers of respondents. Linear regression analysis was used to determine predictors of the dependent variable hemoglobin. Multivariate logistic regression analyses were used to examine the association between the explanatory variables and the outcome variable. Multicollinearity among the explanatory variables was checked using the variance inflation factor (VIF), and variables with VIF less than 2.5 were included. Odds ratios with 95% confidence interval were calculated for each factor in the logistic regression model. Statistical significance was declared if $p$ value was < 0.05.

## Results

The mean age of the women was 23 years and the age range was between 18 and 36 (Table 1). More than 41% of the women had BMI < 20 kg/m$^2$ and only 5% had BMI > 25 kg/m$^2$. Based on the wealth index category more than 39% were poor; 47% had some level of food insecurity, with almost 12% severely food insecure. The majority (62%) of the women were illiterate. Among the literate women, 11 had completed high school and four had some high school education.

The median (IQR) hemoglobin, after adjusting for altitude, was 132 (123, 139) g/L. Among the women, 19% were anemic with 12.5% mild and 6.6% moderately anemic, but none were severely anemic (Fig 1). Based on ferritin and hemoglobin, 7% had iron deficiency anemia, 31% were iron deficient (ferritin < 15 µg/L), and 2% had iron overload (ferritin > 150 µg/L) (Fig 2). Median ferritin, TfR, and hepcidin concentrations were 28 (12, 56) µg/L, 3.5 (2, 5) mg/L, and 7.2 (4, 11) µg/L respectively (Table 1). Of the women 8% had functional iron deficit (TfR ≥ 8.3 mg/L), 6% had acute inflammation (CRP ≥ 5mg/L) and 13% had chronic inflammation (AGP > 1g/L). Tissue iron deficiency (body iron < 0 mg/kg) was found in 16% of the women. Three quarters (78%) of the women were zinc deficient, out of which more than 16% were anemic. Although plasma selenium is highly affected by geographic location, the value for healthy adults has been suggested to vary from 0.5–2.5 µmol/L [29]. None of the women had plasma selenium concentration below 2 µmol/L, and the range was between 2.0 and 5.0 µmol/L with 79% above 2.5 µmol/L.

Correlation coefficients between hemoglobin, iron biomarkers, inflammation indicators and selected minerals showed that Hb was positively associated with plasma iron (r = 0.33, p < 0.001) and plasma zinc (r = 0.23, p = 0.005) and negatively associated with TfR (r = -0.19, p = 0.034) and AGP (r = -0.23, p = 0.004). Ferritin was positively associated with plasma iron (r = 0.32, p < 0.001) and hepcidin (r = 0.48, p < 0.001) and negatively associated with TfR (r = -0.24, p = 0.007). Positive association was observed between CRP and AGP (r = 0.34, p < 0.001). Both CRP and AGP were weakly associated with plasma iron (r = -0.17, p = 0.04)

**Table 1. Demographic and Socioeconomic characteristics and concentrations of biomarkers of iron status, inflammatory markers, and zinc and selenium of lactating women in Sidama zone, southern Ethiopia.**

| Variable | Mean (SD), median (IQR), % |
|---|---|
| Age, year | 23.3 (4.2) |
| • ≤ 25 | 75.7 |
| • 26–35 | 23 |
| • ≥ 25 | 1.3 |
| Household size | 5.7 (2.2) |
| • ≤ 4 | 38.5 |
| • 5–7 | 41.2 |
| • ≥ 8 | 20.3 |
| Number of children | 3.1(1.9) |
| • 1–2 | 44.1 |
| • 3–5 | 44.1 |
| • > 5 | 11.8 |
| MUAC, cm | 24.4 (2.4) |
| BMI, kg/m$^2$ | 20.7 (2.3) |
| Wealth index | |
| • Poorest | 19.7 |
| • Poorer | 19.7 |
| • Middle | 20.4 |
| • Richer | 20.4 |
| • Richest | 19.7 |
| Household food insecurity | |
| • Food secure | 52.6 |
| • Mild food insecure | 12.5 |
| • Moderate food insecure | 23.0 |
| • Severe food insecure | 11.8 |
| Women education | |
| • Illiterate | 61.8 |
| • Literate | 38.2 |
| Hemoglobin, g/L | 131.5 (123, 139) |
| Ferritin, μg/L | 27.6 (12, 56) |
| TfR, mg/L | 3.5 (2, 5) |
| Plasma iron, μmol/L | 14.7 (11, 20) |
| Hepcidin, μg/L | 7.2 (4, 11) |
| CRP, mg/L | 0.8 (0.4, 1.9) |
| AGP, g/L | 0.7 (0.5, 0.8) |
| Plasma zinc μmol/L | 9.5 (8, 11) |
| Plasma selenium μmol/L | 3 (2.6, 3.4) |

and (r = -0.16, p = 0.048) respectively. Plasma zinc was only associated with Hb, and selenium didn't correlate significantly with any of the biomarkers.

Results from a best-fitting multiple linear regression model with eight predictor variables for Hb concentration are presented in Table 2. The variables included were iron biomarkers, inflammation indicators, and selected minerals. Among these, TfR, hepcidin, plasma iron, plasma zinc, and AGP were significant predictors of hemoglobin concentration. In this multiple regression model for hemoglobin, plasma iron contributed the largest proportion of variance, followed by plasma zinc, AGP, and TfR. Hepcidin contributed a small but significant

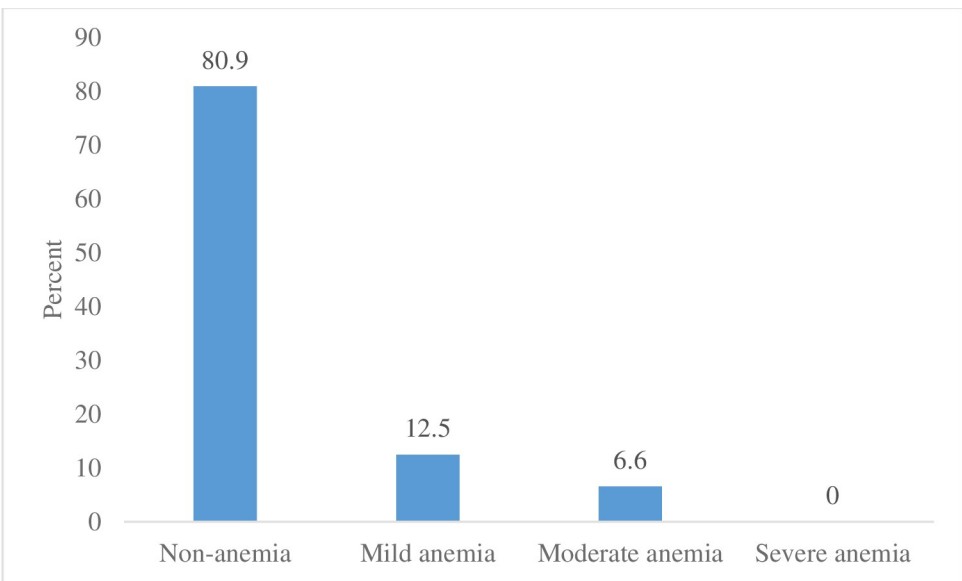

**Fig 1. Prevalence of anemia in lactating women in southern Ethiopia.**

proportion to the hemoglobin variance. The contributions of ferritin, CRP, and selenium were not significant. The regression model explained 30.6% of the variance in hemoglobin concentrations, but when plasma selenium was removed from the regression the variance explained increased to 31.2%. The variance explained decreased to 27% when zinc was removed from the multiple regression model. The squared semi partial correlation indicated that zinc and AGP explained nearly 4 times more of the variance than ferritin concentration (Table 2).

To examine determinant factors of anemia, a bivariate logistic regression analysis was performed. In this analysis all of the variables included in the multiple linear regression were

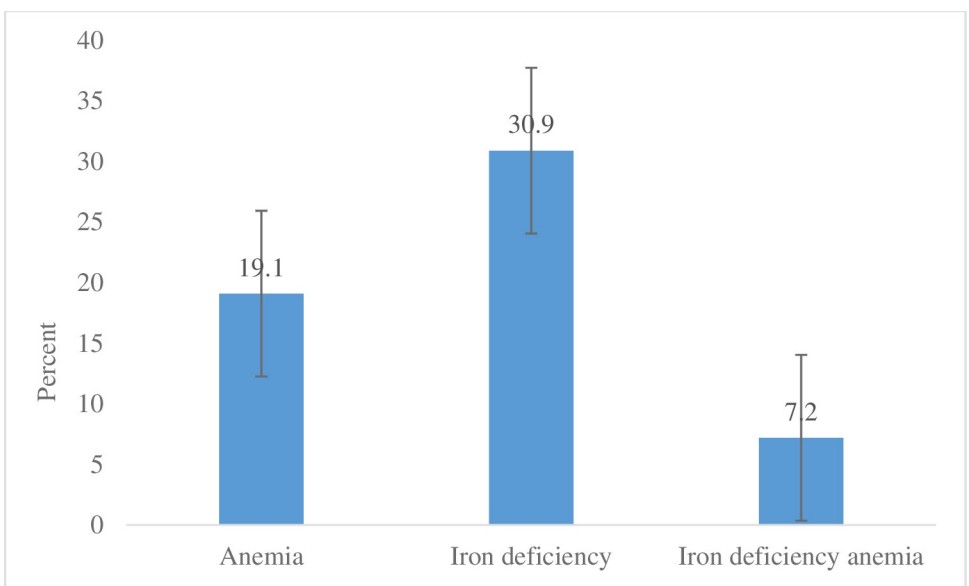

**Fig 2. Prevalence of anemia, iron deficiency and iron deficiency anemia in lactating women in southern Ethiopia.**

**Table 2. Multiple regression analysis for variables predicting hemoglobin concentration in lactating women in Sidama zone, southern Ethiopia.**

| Variable[d] | Coefficient (95% CI) | p value | Squared semi partial correlation | p value |
|---|---|---|---|---|
| Ferritin | -0.08 (-0.21, 0.061) | 0.413 | 0.012 | 0.180 |
| TfR[a] | -1.78 (-2.33, -1.23) | 0.001 | 0.043 | 0.020 |
| Hepcidin | -0.03 (-0.051, -0.013) | 0.032 | 0.029 | 0.036 |
| Plasma iron | 1.68 (0.87, 2.21) | 0.002 | 0.083 | 0.001 |
| Plasma zinc | 1.16 (0.86, 2.23) | 0.011 | 0.047 | 0.008 |
| Plasma selenium | 0.011 (-0.065, 0.087) | 0.771 | 0.003 | 0.543 |
| CRP[b] | -1.01 (-3.03, 1.16) | 0.231 | 0.011 | 0.215 |
| AGP[c] | -0.304 (-0.51, -0.089) | 0.001 | 0.047 | 0.008 |

Adjusted R square = 30.6

[a] Transferrin receptor

[b] C reactive protein

[c] α-1-acid glycoprotein

[d] All variables are $\log_{10}$ transformed

included in addition to the socio-demographic variables. The variables that showed significant association were fitted in a multivariate logistic regression analysis. In multivariate analysis, TfR, AGP, MUAC, and education were significant determinant factors of anemia. Lactating women with sufficient functional iron (TfR < 8.3 mg/L) were 96.5% less likely to be anemic (OR = 0.035; 95% CI 0.006, 0.198). Similarly, women without chronic inflammation (AGP < 1 g/L) were 88.6% less likely to be anemic (OR = 0.114; 95% CI 0.025, 0.522). Women with small muscle mass (MUAC < 22 cm) were 6.78 times (95% CI 1.56, 29.45) and illiterate women were 4.94 times (95% CI 1.06, 23.01) more likely to be anemic (Table 3).

## Discussion

In this study, iron deficiency was fairly high (31%) and prevalence of anemia was 19%, which was slightly lower than the 21% previous reported for women of reproductive age in the same study area [30]. However, iron deficiency anemia was only 7%. The etiology of anemia is complex because of multiple contributors, including nutritional and non-nutritional factors that can be directly or indirectly related to each other. The suggestion has been that 50% of anemia is caused by iron deficiency [21] but this value may not be consistently applicable to women in various stages of the life cycle and may not account for the contribution of infection to anemia in different settings [6].

In the current study, plasma iron, AGP, TfR, hepcidin and plasma zinc were significant predictors of maternal anemia. Plasma iron was the major contributor to the variance followed by AGP and plasma zinc. Ferritin was not a significant predictor, and even its contribution to the variance was small compared to the other variables. Iron is needed for various biological processes and body iron balance is maintained by complex regulatory mechanisms [31]. For instance, iron absorption and iron release from cells is regulated by a hepatic peptide hormone hepcidin, which also regulates plasma iron concentration by controlling recycling or storing iron [32]. However, excess production of hepcidin was associated with iron restricted anemia including anemia associated with inflammation [33]. In fact, inflammation is one of the major stimuli regulating hepcidin transcription other than plasma iron concentration [33].

Although multiple iron biomarkers predicted maternal anemia, AGP and TfR were the biomarkers associated with hemoglobin in the logistic regression model. Absence of chronic

**Table 3. Multivariate logistic regression analysis with anemia as the outcome variable for lactating women in Sidama zone, southern Ethiopia.**

| Variable | OR (95% CI) | P value |
|---|---|---|
| TfR (mg/L) | | |
| < 8.3 | 0.035 (0.006, 0.198) | 0.001 |
| ≥ 8.3 | r* | |
| AGP (g/L) | | |
| ≤ 1 | 0.114 (0.025, 0.522) | 0.005 |
| > 1 | r | |
| MUAC (cm) | | |
| < 22 | 6.78 (1.56, 29.45) | 0.011 |
| ≥ 22 | r | |
| Maternal education | | |
| Illiterate | 4.94 (1.06, 23.01) | 0.042 |
| Literate | r | |
| Maternal age (Years) | | |
| ≤ 21 | 2.93 (0.55, 15.66) | 0.209 |
| > 21 | r | |
| Plasma zinc μmol/L | | |
| < 10.7 | 1.09 (0.25, 4.86) | 0.908 |
| ≥ 10.7 | r | |
| Gravidity | | |
| 1–3 | 0.48 (0.087, 2.603) | 0.391 |
| 4–8 | r | |

*r—reference

inflammation and having sufficient functional iron were associated with a decreased risk of anemia independent of low ferritin known to contribute to low hemoglobin [34].

Iron biomarkers need to be adjusted for inflammation before statistical analysis because iron status is affected by inflammation. Malaria and intestinal parasitic infestation are among the major public health problems in the study area that give rise to increased anemia as well as inflammation [35, 36]. Malaria can cause anemia through the destruction of erythrocytes, and parasites can increase risk of infectious diseases and inflammation [37, 38]. Furthermore, in the presence of anemia of inflammation, low serum iron concentration is very common [39]. In this study there were nine women with acute inflammation (incubation stage) and 19 women with chronic inflammation (late convalescence). Twelve women who were anemic had high AGP or CRP. However, AGP was among the major predictors of anemia and second only to TfR as a biochemical determinant factor of anemia. CRP didn't show significant association or prediction of anemia. This could be because CRP concentration decreases rapidly in convalescence and AGP increases slowly and remains elevated for some time in convalescence or chronic infection [40, 41]. In pregnant women in the same study area, CRP was a major predictor of anemia [35]. Results from the BRINDA project suggest that the extent of the contribution of iron deficiency to anemia depends on the infection burden [6].

Other studies have shown that nutrients such as zinc are major contributors to anemia independent of ferritin, which is one of the major iron biomarkers [7]. Many (78%) of the lactating women who participated in our study had low plasma zinc, and the study region has a

long history of zinc deficiency. Both inadequate intake of dietary zinc and their high phytate maize based diet, contribute to the problem [42, 43]. Although it has not been common to assess zinc deficiency in relation to anemia, there are multiple mechanisms by which zinc could contribute to anemia. More than 300 enzymes in our body are zinc-dependent, including polymerases needed for the synthesis of DNA as well as amnolevulinic acid dehydrase for synthesis of heme. Also of note is zinc's role in a zinc finger transcription factor which plays a major role in erythropoiesis [44, 45]. A study in Cambodian children and women suggested that anemia should not be explained only by iron and hemoglobin disorders but also by other nutrients such as zinc and folate, as well as parasitic infestation [46]. Another study reported that serum zinc concentration was significantly lower in anemic women than in their counterparts, and zinc deficiency aggravated iron deficiency anemia [47].

The risk of anemia decreased with increased muscle mass as measured by MUAC. In poor resource settings, MUAC was a strong predictor of underweight and anemia in women of reproductive age [48]. Low MUAC could indicate undernutrition, and undernutrition is directly related to food insecurity. Mothers in food insecure households are more likely to be undernourished than mothers in food secure households [49]. However, in our previous study food insecurity was not a significant predictor of low hemoglobin in women of reproductive age [30].

In Ethiopia majority of the population depends on subsistence farming, food insecurity is highly prevalent and most women historically have had limited education. In the current study, education was a significant determinant factor of anemia perhaps because those who are better educated are more likely to be employed and have better access to nutritious food and improved health services than the less educated [50].

In conclusion, a combination of factors including nutritional and non-nutritional factors contribute to anemia. The extent of the contribution of various factors depends on a number of conditions. For instance, in settings where the burden of infection is high, the contribution of nutrients in the etiology of anemia could be undermined. Hence in the effort to minimize prevalence of anemia, it may well be necessary to reduce contributors to infection and inflammation before measures such as nutrient supplementation or fortification are taken.

## Supporting information

**S1 Data.**
(PDF)

## Acknowledgments

We thank the study participants who took part in this study and Mr. Keneni Fufa who collected the blood samples.

## Author Contributions

**Conceptualization:** Tafere Gebreegziabher, Barbara J. Stoecker.

**Data curation:** Tafere Gebreegziabher, Barbara J. Stoecker.

**Formal analysis:** Tafere Gebreegziabher, Taylor Roice, Barbara J. Stoecker.

**Funding acquisition:** Tafere Gebreegziabher, Barbara J. Stoecker.

**Investigation:** Tafere Gebreegziabher.

**Methodology:** Tafere Gebreegziabher, Taylor Roice, Barbara J. Stoecker.

**Project administration:** Tafere Gebreegziabher, Barbara J. Stoecker.

**Supervision:** Barbara J. Stoecker.

**Visualization:** Taylor Roice.

**Writing – original draft:** Tafere Gebreegziabher.

**Writing – review & editing:** Tafere Gebreegziabher, Taylor Roice, Barbara J. Stoecker.

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
