## [Decision Letter · Decision Letter 0]

1 Sep 2020

PONE-D-20-19809

Chronic inflammation was a major predictor and determinant factor of anemia in lactating women in Sidama zone southern Ethiopia: a cross-sectional study

PLOS ONE

Dear Dr. Gegreegziabher,

Thank you for submitting your manuscript to PLOS ONE. After careful consideration, we feel that it has merit but does not fully meet PLOS ONE’s publication criteria as it currently stands. Therefore, we invite you to submit a revised version of the manuscript that addresses the points raised during the review process.

We look forward to receiving your revised manuscript.

Kind regards,

Gary Kupfer

Academic Editor

PLOS ONE

Journal Requirements:

2. In your Methods section, please provide additional information about the participant recruitment method and the demographic details of your participants. Please ensure you have provided sufficient details to replicate the analyses such as: a) the recruitment date range (month and year), b) a description of any inclusion/exclusion criteria that were applied to participant recruitment, c) a table of relevant demographic details, d) a statement as to whether your sample can be considered representative of a larger population, e) a description of how participants were recruited, and f) descriptions of where participants were recruited and where the research took place. Moreover, please clarify how the wealth index was calculated.

3. Please correct your reference to "p=0.000" to "p<0.001" or as similarly appropriate, as p values cannot equal zero.

4. Thank you for stating the following in the Financial Disclosure section:

"The research was funded by the USDA Multistate Project, W-3002 to BJS; Nestlé Foundation  to TG.

We note that you received funding from a commercial source: Nestle.

Reviewers' comments:

Reviewer's Responses to Questions

**Comments to the Author**

1. Is the manuscript technically sound, and do the data support the conclusions?

Reviewer #1: Yes

Reviewer #2: Yes

2. Has the statistical analysis been performed appropriately and rigorously? 

Reviewer #1: Yes

Reviewer #2: Yes

3. Have the authors made all data underlying the findings in their manuscript fully available?

Reviewer #1: Yes

Reviewer #2: Yes

4. Is the manuscript presented in an intelligible fashion and written in standard English?

Reviewer #1: Yes

Reviewer #2: Yes

5. Review Comments to the Author

Reviewer #1: - The main claims for this paper are that anemia in the geographic area of study is associated with AGP, a marker of chronic inflammation, and the authors propose that this chronic inflammation is from infection from malaria/parasites. The authors analyzed the data in such a way to suggest that chronic inflammation is a top factor in patients having anemia which seems to be the novel contribution of this manuscript. The authors suggest that interventions to decrease anemia might need to include treating infection before nutrition supplementation to decrease this chronic inflammation.

- Previous literature (https://www.ncbi.nlm.nih.gov/pmc/articles/PMC6767796/) has described the association between malaria with anemia and laboratory markers of inflammation, but this previous literature has not performed the type of analysis performed in this paper providing some possible ranking on the contribution of various factors to anemia in areas with high rates of malaria. The novel information provided by this paper seems to be that chronic inflammation plays a leading role in anemia in this geographic area with high rates of malaria. On one hand this could be practical/impactful information to guide policy on how to improve anemia in malaria infected areas by focusing first on treating the chronic inflammation by treating the malaria. On the other hand, there already is data to suggest that iron supplementation in endemic areas for malaria could prove harmful (https://www.ncbi.nlm.nih.gov/pmc/articles/PMC3124144/#B116) which would seem to suggest that treating malaria should be done before supplementing with iron, which is the same policy message suggested by the authors of this paper.

- The authors should consider placing their study results in the context of the literature describing the interplay between anemia and malaria and inflammation (https://www.ncbi.nlm.nih.gov/pmc/articles/PMC3124144/#B116, https://www.ncbi.nlm.nih.gov/pmc/articles/PMC3124144/#B116) to give more context of how their finding of inflammatory markers with chronic inflammation playing a large role in anemia fits in with the context of past data showing that supplementation with iron in malaria endemic areas could be harmful.

- The authors do a good job of not overstating their claims based on their data analysis, by only mildly suggesting that infection should be treated before nutritional supplementation for anemia.

- Can the authors provide specific data on the proportion of patients with anemia who also have markers of chronic inflammation?

Reviewer #2: The manuscript by Gebreegziabher purports to analyze anemia in Ethiopian women and factors promoting the disease state. Focus was on lactating women in areas prone to infection, esp malaria. Surprisingly, the fraction of cases of anemia attributable to iron deficiency in isolation was small, and the authors ascribe a large percentage to chronic inflammation. The documentation and correlation of low fraction of iron def and high degree of chronic inflammation is interesting and helpful in the way public health authorities may think about health in underserved communities.

The paper would be helped by a more careful English edit

Are the authors able to report B12, folate?

6. PLOS authors have the option to publish the peer review history of their article (what does this mean?). If published, this will include your full peer review and any attached files.

Reviewer #1: No

Reviewer #2: No

---

## [Author Response · Author response to Decision Letter 0]

14 Sep 2020

 - Formatted according to PLOS ONE style

2. In your Methods section, please provide additional information about the participant recruitment method and the demographic details of your participants. Please ensure you have provided sufficient details to replicate the analyses such as: a) the recruitment date range (month and year), b) a description of any inclusion/exclusion criteria that were applied to participant recruitment, c) a table of relevant demographic details, d) a statement as to whether your sample can be considered representative of a larger population, e) a description of how participants were recruited, and f) descriptions of where participants were recruited and where the research took place. Moreover, please clarify how the wealth index was calculated.

 a) The recruitment date range June to August, 2013 in line 98. 

b) Inclusion criteria were included in lines 99 – 101. Eligibility criteria were: the woman must be 18 years of age or older, must be lactating six months after delivery and must have no history of illness.

c) Demographic characteristics (age, household size and number of children) are included to table 1.

d) A statement included in line 101 – 102. Because all women agreed to join the study, we consider the study representative of the rural Sidama population.

e) A description of where participants were recruited were given in line 97 – 98 and 102 – 103. We conducted a cross-sectional study of lactating Ethiopian women from eight randomly selected rural villages in Sidama zone, southern Ethiopia. Women came to their local community health post for the data collections.

f) A description of where the research took place was given on line number 97 - 98. Please refer to ‘e’ above.

- A statement on how wealth index was calculated was given on line number 120 – 121. Principal component analysis (PCA) was applied to compute wealth index, and the score was used to divide the participants into five quintiles.

3. Please correct your reference to "p=0.000" to "p<0.001" or as similarly appropriate, as p values cannot equal zero.

 - “p = 0.000 is changed to p < 0.001”

4. Thank you for stating the following in the Financial Disclosure section:

"The research was funded by the USDA Multistate Project, W-3002 to BJS; Nestlé Foundation to TG.

We note that you received funding from a commercial source: Nestle.

Thank you for the comment. Nestlé Foundation is not a commercial organization. Please see the link for Nestlé Foundation http://www.nestlefoundation.org/e/

The Nestle Foundation has a review board of esteemed scientists and has a history of supporting nutrition projects in developing countries.

5. Review Comments to the Author

Reviewer #1: - The main claims for this paper are that anemia in the geographic area of study is associated with AGP, a marker of chronic inflammation, and the authors propose that this chronic inflammation is from infection from malaria/parasites. The authors analyzed the data in such a way to suggest that chronic inflammation is a top factor in patients having anemia which seems to be the novel contribution of this manuscript. The authors suggest that interventions to decrease anemia might need to include treating infection before nutrition supplementation to decrease this chronic inflammation.

- Previous literature (https://www.ncbi.nlm.nih.gov/pmc/articles/PMC6767796/) has described the association between malaria with anemia and laboratory markers of inflammation, but this previous literature has not performed the type of analysis performed in this paper providing some possible ranking on the contribution of various factors to anemia in areas with high rates of malaria. The novel information provided by this paper seems to be that chronic inflammation plays a leading role in anemia in this geographic area with high rates of malaria. On one hand this could be practical/impactful information to guide policy on how to improve anemia in malaria infected areas by focusing first on treating the chronic inflammation by treating the malaria. On the other hand, there already is data to suggest that iron supplementation in endemic areas for malaria could prove harmful (https://www.ncbi.nlm.nih.gov/pmc/articles/PMC3124144/#B116) which would seem to suggest that treating malaria should be done before supplementing with iron, which is the same policy message suggested by the authors of this paper.

- The authors should consider placing their study results in the context of the literature describing the interplay between anemia and malaria and inflammation (https://www.ncbi.nlm.nih.gov/pmc/articles/PMC3124144/#B116, https://www.ncbi.nlm.nih.gov/pmc/articles/PMC3124144/#B116) to give more context of how their finding of inflammatory markers with chronic inflammation playing a large role in anemia fits in with the context of past data showing that supplementation with iron in malaria endemic areas could be harmful.

- Reference included (Reference no.38)

- The authors do a good job of not overstating their claims based on their data analysis, by only mildly suggesting that infection should be treated before nutritional supplementation for anemia.

- Thank you for the comment

- Can the authors provide specific data on the proportion of patients with anemia who also have markers of chronic inflammation?

- Based on your suggestion we have included the following sentence in the discussion line 276. ‘12 women who were anemic had high AGP or CRP.

Reviewer #2: The manuscript by Gebreegziabher purports to analyze anemia in Ethiopian women and factors promoting the disease state. Focus was on lactating women in areas prone to infection, esp malaria. Surprisingly, the fraction of cases of anemia attributable to iron deficiency in isolation was small, and the authors ascribe a large percentage to chronic inflammation. The documentation and correlation of low fraction of iron def and high degree of chronic inflammation is interesting and helpful in the way public health authorities may think about health in underserved communities.

The paper would be helped by a more careful English edit

Are the authors able to report B12, folate?

-The English has been carefully edited.

-Thank you for the comment. We did not analyzed B12 and folate.

---

## [Editor Report · Decision Letter 1]

23 Sep 2020

Chronic inflammation was a major predictor and determinant factor of anemia in lactating women in Sidama zone southern Ethiopia: a cross-sectional study

PONE-D-20-19809R1

Dear Dr. Gebreegziabher,

We’re pleased to inform you that your manuscript has been judged scientifically suitable for publication and will be formally accepted for publication once it meets all outstanding technical requirements.

Kind regards,

Gary Kupfer

Academic Editor

PLOS ONE
---

## [Editor Report · Acceptance letter]

25 Sep 2020

PONE-D-20-19809R1 

Chronic inflammation was a major predictor and determinant factor of anemia in lactating women in Sidama zone southern Ethiopia: a cross-sectional study 

Dear Dr. Gebreegziabher:

I'm pleased to inform you that your manuscript has been deemed suitable for publication in PLOS ONE. Congratulations! Your manuscript is now with our production department. 

Kind regards, 

on behalf of

Dr Gary Kupfer 

Academic Editor

PLOS ONE